# Mapping of Microglial Brain Region, Sex and Age Heterogeneity in Obesity

**DOI:** 10.3390/ijms22063141

**Published:** 2021-03-19

**Authors:** Irina V. Milanova, Felipe Correa-da-Silva, Andries Kalsbeek, Chun-Xia Yi

**Affiliations:** 1Department of Endocrinology and Metabolism, Amsterdam University Medical Center (UMC), University of Amsterdam, 1105 AZ Amsterdam, The Netherlands; i.v.milanova@amsterdamumc.nl (I.V.M.); f.correadasilva@amsterdamumc.nl (F.C.-d.-S.); a.kalsbeek@nin.knaw.nl (A.K.); 2Laboratory of Endocrinology, Amsterdam University Medical Center (UMC), University of Amsterdam, Amsterdam Gastroenterology & Metabolism, 1105 AZ Amsterdam, The Netherlands; 3Netherlands Institute for Neuroscience, Royal Netherlands Academy of Arts and Sciences, 1105 BA Amsterdam, The Netherlands

**Keywords:** microglia, obesity, brain region, heterogeneity, hypothalamus, hippocampus

## Abstract

The prevalence of obesity has increased rapidly in recent years and has put a huge burden on healthcare worldwide. Obesity is associated with an increased risk for many comorbidities, such as cardiovascular diseases, type 2 diabetes and hypertension. The hypothalamus is a key brain region involved in the regulation of food intake and energy expenditure. Research on experimental animals has shown neuronal loss, as well as microglial activation in the hypothalamus, due to dietary-induced obesity. Microglia, the resident immune cells in the brain, are responsible for maintaining the brain homeostasis and, thus, providing an optimal environment for neuronal function. Interestingly, in obesity, microglial cells not only get activated in the hypothalamus but in other brain regions as well. Obesity is also highly associated with changes in hippocampal function, which could ultimately result in cognitive decline and dementia. Moreover, changes have also been reported in the striatum and cortex. Microglial heterogeneity is still poorly understood, not only in the context of brain region but, also, age and sex. This review will provide an overview of the currently available data on the phenotypic differences of microglial innate immunity in obesity, dependent on brain region, sex and age.

## 1. Introduction

Microglial cells, the resident innate immune cells in the central nervous system (CNS), have risen as a key player in many pathologies, including metabolic syndrome, neurodegenerative and cardiovascular conditions. Their main role is surveillance of the brain microenvironment. Resident microglial cells continuously survey the CNS to detect pathological changes. When responding to an immune challenge, they quickly adapt from a resting into an activated state. Once activated, microglia secrete cytokines, thus mediating the innate immune response. Another key microglial function, important for maintaining brain homeostasis, is phagocytosis. By phagocyting cellular debris and dying cells, microglia prevent the release of proinflammatory and neurotoxic molecules in the brain environment [1].

Experimental studies in rodents have shown that a hypercaloric diet is a potent activator of microglial cells [2,3,4]. Throughout the years, research has focused on characterizing microglial cells based on their two types of activation states—the so-called M1 activation, characteristic of a proinflammatory response and cytokine production, and M2 polarization, responsible for maintaining an anti-inflammatory response [5]. However, rising evidence suggests a much broader spectrum of phenotypic differences, both in the context of different pathologies, as well as with regards to brain region specificity. 

The prevalence of obesity tripled between 1975 and 2016, with 13% of the world’s adult population being obese in 2016 (WHO, 2016). These numbers are projected to increase further, highlighting it as one of the main health problems worldwide, due to an increased risk of diabetes, hypertension and cardiovascular disease. Although the underlying cause of this pathology is clear, i.e., excessive food intake and reduced energy expenditure, these morbid statistics clearly show that our understanding of the pathogenesis of obesity is still incomplete. The key brain region responsible for maintaining homeostasis is the hypothalamus, but clearly, energy homeostasis is not maintained during obesity. The discovery of leptin, a hormone produced by adipose tissue and the product of the obese (ob) gene, illustrated the involvement of the brain in the energy balance by its pronounced effects on hypothalamic activity [6] but, as for now, has not provided the necessary answers. Moreover, obesity is also highly associated with changes in the hippocampal function, which could ultimately result in cognitive decline and dementia.

This review aims to assess the available studies evaluating the brain region, sex and age heterogeneity in the microglial function in obesity. The review is separated according to the different brain regions evaluated, followed by a section evaluating the effect of overnutrition on the microglial function in the pre- and neonatal stages of life, taking into account sex differences whenever available. The majority of the studies included in this review were performed on experimental rodents, unless stated otherwise (e.g., humans). 

## 2. Hypothalamus

The hypothalamus is the brain hub responsible for the regulation of key metabolic and endocrine functions, as well as the regulation of body temperature, sleep/wake rhythms and feeding behavior, maintaining a strict control of the balance between food intake and energy expenditure. Within the mediobasal hypothalamus (MBH), the arcuate nucleus (ARC) is the key region maintaining metabolic homeostasis, containing both an anorexigenic proopiomelanocortin (POMC) plus cocaine- and amphetamine-regulated transcript (CART) and an orexigenic neuropeptide Y (NPY) plus agouti-related peptide (AgRP) neuronal population. The ARC is the primary center of the brain involved in the sensing of nutrients. In addition to nutrient sensing, the anorexigenic and orexigenic neurons of the ARC respond to peripheral metabolic hormones like insulin, leptin and ghrelin. Via their projections to secondary neurons in the dorsomedial, paraventricular and lateral hypothalamus, energy homeostasis is maintained [7]. Therefore, the hypothalamic function in obesity has been studied vastly, both in rodent models using diets with high-fat contents (45–60% kcal fat), i.e., dietary-induced obesity (DIO), as well as genetic models representative of this pathology. 

It has been shown that the depletion of microglial cells leads to reduced food intake, total body fat and weight gain in mice fed an obesogenic diet, with no changes observed in the same animals when fed regular chow [2,8]. This suggests that microglia could be a key player in the complex hypothalamic regulation of the energy balance by sustaining an elevated caloric intake, which will ultimately lead to increased adiposity and body weight. In rodents fed a regular chow diet, microglia show a ramified morphology, characteristic for the resting state. However, a high-fat diet (HFD) feeding is associated with morphological changes, indicative of microglial activation [9]. Moreover, unlike inflammation in peripheral tissues, which appears as a consequence of obesity, hypothalamic inflammation is observed in rats and mice within one day following HFD feeding, which is prior to any weight gain [10]. This suggests that hypothalamic microglia are among the first responders to HFD feeding.

Ionized calcium-binding adapter molecule 1 (Iba-1) is a microglia-specific protein. The prolonged feeding of wild-type (WT) mice (C57BL6 background) with a HFD increases the Iba-1+ microglial activation in the ARC, as indicated by the increased cell number and/or morphological changes (increased cell soma size and thicker processes) [4,11,12,13,14,15,16]. Another hypothalamic nucleus important for metabolic homeostasis is the paraventricular nucleus (PVN), as it controls both the neuroendocrine and autonomic output of the hypothalamus [17,18]. The consumption of an obesogenic diet resulted in an increased cluster of differentiation molecule 11b (Cd11b) mRNA expression (Cd11b being another microglial marker), as well as increased Iba-1+ microglial cell numbers in the PVN, suggesting that the effect of obesity on microglial cells is not restricted to the ARC [11,19]. In addition, not only the amount of calories but, also, diet composition seem to influence microglial inflammation, as chronic feeding with a cafeteria diet (chow diet plus 12% sucrose solution, mimicking soda consumption) did not change the Iba-1+ cell density in the ARC, suggesting a role of dietary fat in the microglial inflammatory response [20]. The reversal from HFD to chow diet reduced the microglial activation, resulting in a cell activation status comparable to that of the control animals [14].

Interestingly, it seems that microglial cells in the MBH change their activity state according to the sleep/wake cycle, since during the active period of the animal, the Iba-1+ cell number and projections are increased relative to the inactive period of the animal [21]. This rhythmic change was observed in rodents (mice and rats) fed a control diet but was absent in DIO animals that showed constantly increased microglial cell numbers and projections. Similar changes were observed in tumor necrosis factor-alpha (TNF-a) secretion, a cytokine released by microglia to initiate a proinflammatory response [21]. The continuously elevated cytokine production could be a possible explanation for the observed neuroinflammation during obesity. 

Microglial immunity is also affected by age, as both HFD- and control diet-fed aged animals (12 months old) have an increased microglial immunoreactivity compared to young animals (six months old) [22], but an obesogenic diet further enhances this aging-induced activation. Some controversy exists with regards to the short-term acute effects of HFD on microglial activation in the hypothalamus. One day of HFD feeding has been shown to lead to microglial activation (as indicated by the increased cell number and soma size), compared to chow feeding [23], but other studies showed no effect after three days or two weeks of HFD feeding on the microglial innate immunity [24,25]. These results are suggestive of an acute microglial response, followed by a transient period. 

As mentioned earlier, leptin is as a key endocrine hormone involved in the CNS control and regulation of energy homeostasis [6,26]. Interestingly, leptin signaling might be crucial for microglial innate immunity. Depleting the leptin receptor specifically in microglial cells has been shown to downregulate microglial cluster of differentiation 68 (CD68, a lysosomal protein highly expressed in activated microglia)-positive phagosomes, suggesting an impairment of the microglial phagocytic capacity [27]. This has been confirmed in genetic models with whole body-disturbed leptin signaling (db/db and ob/ob), suggesting that leptin deficiency and leptin receptor mutations can lead to microglial under activation and impaired phagocytosis [3]. Another endocrine hormone produced by the gut is ghrelin, a peptide stimulating food intake and decreasing energy expenditure, thus exerting a physiological effect opposite to that of leptin. A single dose of ghrelin prior to the start of an obesogenic diet reduced the number of Iba-1+ microglial cells in the hypothalamus after one day of HFD feeding [23]. Whether ghrelin also has a long-term protective effect on microglial activation in DIO is unknown. 

Lipoprotein lipase (Lpl) is a key enzyme involved in catalyzing the breakdown of triglyceride-rich lipoproteins and, thus, allowing the cellular uptake of lipids. A recent study has found that, in a hypercaloric environment, LPL is also important for the microglial innate immune function. Depletion of the Lpl gene specifically from microglial cells resulted in decreased microglial cell numbers and activation, as well as decreased phagocytic capacity in the MBH of animals fed an obesogenic diet [28]. Moreover, this decrease in innate immunity was specific for an obesogenic environment, as no differences were observed between chow-fed WT and microglial Lpl knockdown animals [28]. 

The study of neuroinflammation in obesity has focused on the key functions of microglial immunity. A study evaluating the long-term effects of DIO on the ubiquitin/proteasome system showed that Cd11b+ microglial cells in the ARC express ubiquitin and p62, two key proteins involved in the regulation and function of this system [29]. Long-, but not short-term HFD exposure, increased the amount of nonubiquinated proteins, suggesting that one of the possible mechanisms in the development of neuroinflammation is impaired protein ubiquitination. Progranulin, a glycoprotein modulating wound healing and inflammation, has been shown to colocalize with Iba-1+ microglial cells in the ARC and the median eminence (ME) of control-fed animals and is elevated in HFD-fed mice [30]. HFD feeding induced an IgG accumulation in Iba-1+ microglial cells, indicating that a major portion of HFD-induced IgG may be localized in microglia. By comparison, there is no IgG colocalizing within GFAP+ astrocytes. Moreover, the IgG subclass IgG1, but not IgG2 or IgG3, contributed to the observed microglial IgG colocalization in the ARC [31]. As each subclass is known to show differences in antigen binding, immune complex formation and complement activation, this suggests a potential mechanism for the microglial immune response in DIO in the ARC [32].

In view of the above, restraining the microglial inflammatory activation could be a potential regulator of obesity-induced neuroinflammation. Indeed, deletion of the inhibitor of the NF-kB kinase (IKKb) in microglial cells led to a decrease in HFD-induced microgliosis, indicating that microglial NF-kB-dependent signaling is necessary for the HFD-induced microglial innate immune response [8]. Another study showed that the global ablation of 4-1BB (also known as CD137, a protein with a costimulatory and inflammatory functions) in mice leads to downregulation of the microglial activation markers (Iba-1 and Cd11b) in HFD mice, compared to obese WT controls [33]. Moreover, the increased Iba-1+ microglial cell number in the hypothalamus induced by an obesogenic diet was further elevated by ablation of the interferon regulatory factor 2 binding protein 2 (Irf2bp2) [34]. Irf2bp2 is a key regulator of the inflammatory response in macrophages and microglial cells by affecting the polarization [35]; thus, these data indicate a possible neuroprotective role of Irf2bp2 in obesity. 

Many compounds have shown promising effects as modulators of microglia-induced neuroinflammation. Quercetin (a naturally occurring flavonoid known to protect against oxidative stress and inflammation) has been shown to reduce obesity-induced inflammation in the hypothalamus by inhibition of the microglial inflammatory response via a reduced expression of key cytokine genes (TNF-a, Il-1b and MCP-1) and an overall decrease in the activation status [36]. The chronic administration of abscisic acid (ABA) reduced HFD-induced microglial activation and TNF-a production in the hypothalamus [37]. Canagliflozin—an inhibitor of the sodium glucose cotransporter 2—has been shown to reverse the HFD-induced increase in the Iba-1+ cell number, suggesting a preventive effect on neuroinflammation [38]. The loss of Sirt-1 in myeloid cells has been shown to further increase microglial activation in the hypothalamus of HFD-fed animals [39], probably due to the inhibitory effect of Sirt-1 on microglial inflammatory cytokine production [40]. AraC (cytarabine) —an antimitotic drug—has been shown to decrease the Iba-1+ microglial cell number and TNF-a expression, thus blunting the microglial activation [41]. Moreover, it has been shown that liraglutide, a glucagon-like peptide-1 (GLP-1) derivative originally developed for the treatment of type 2 diabetes, has a neuroprotective effect by rescuing HFD-induced microgliosis in the ARC of mice [42]. The absence of growth hormone (GH) signaling also seems to have a neuroprotective effect in HFD-fed mice, as lacking the GH receptor blunted the Iba1 protein expression increase [43]. It is worth noting that these effects could improve the DIO-induced microgliosis either directly, acting upon microglia, or indirectly by treating obesity and, thus, reducing the microglial activation. More research, such as pair-feeding experiments, should be performed to understand the mechanisms of action of these compounds as microglial regulators and to evaluate their safety and efficacy for use in obesity, as well as other diseases. 

Another suitable regulator of microglial activation is physical activity. Moderate exercise seems to have a protective effect in obesity by reducing obesity-related neuroinflammation. Low-density lipoprotein receptor (Ldlr) knockout mice (a genetic model for metabolic syndrome) fed an obesogenic diet showed increased microglial activation specifically in the ARC, which was abolished by regular and moderate treadmill running [44]. Dim light exposure at night (dLAN) increased the Iba-1+ cell number in control animals, but no effect was found in HFD-fed animals [16].

Obesity also seems to predispose those affected towards a worsened outcome in traumatic brain injuries (TBI). TBI in obese animals has been shown to increase the microglial activation in the hypothalamus, primarily in the ventromedial hypothalamic nucleus (VMH), when compared to obesity alone, while there is no effect of TBI on microglial activation in lean animals [45].

Interestingly, many studies have shown sex-specific differences in the susceptibility to obesity and microglial activation. The worsened outcome following TBI mentioned above was found in male mice, but obese female mice showed similar numbers of microglial activation before and after TBI. Moreover, when compared to obese male TBI mice, an overall lower microglial activity was found in females [45]. A possible explanation for this observation is a sex-specific resistance in females to DIO-induced microglial activation, mediated via the fractalkine receptor chemokine (C-X3-C motif) receptor 1 and its highly selective ligand Cx3Cl1, known as the Cx3Cl1–Cx3Cr1 axis [46]. The Cx3Cl1–Cx3Cr1 axis mediates chemotaxis and the adhesion of immune cells via the secretion of Cx3Cl1 by neurons. Hypothalamic microglia from HFD-fed males showed increased immune activation (as indicated by the increases in Il-1b and IKKb) when compared to controls, while no such changes were observed in females. The evaluation of the Cx3Cl1–Cx3Cr1 pathway showed an increase in Cx3Cl1 in HFD-fed females, with no differences at the receptor levels of Cx3Cr1, while both were decreased in obese males, suggesting a sex-specific protective effect. This was further highlighted by obese females with a microglia-specific deletion of the fractalkine receptor Cx3Cr1, which showed a rapid expansion of microglial activation in the MBH, comparable to HFD-fed males [46]. This suggests that fractalkine exerts an inhibitory effect on microglial activation in obesity. This idea was confirmed in fat-1 transgenic mice, known to be resistant to DIO, which showed decreased microglial and cytokine gene expression but increased Cx3Cl1 expression [47]. Another study, investigating the effects of arachidonic acid on obesity, showed an increased Iba-1+ microglial number in the ARC of male mice, as well as increased mRNA levels for Iba1, TNF-a, Il6 and TLR4, suggesting microglial activation, which was further worsened by a treatment with arachidonic acid [48]. Interestingly, no such changes were observed in females. 

Microglial activation has also been found in obese humans. A histological analysis of brain tissue from obese humans (body mass index (BMI) > 30) showed increased microglial activation (as seen in morphological changes in Iba1+ cells, enlarged cell soma and shortened processes) compared to control patients (BMI < 25) [24]. Remarkably, this increased activation was exclusively observed in the hypothalamic area adjacent to the third ventricle. Moreover, an RNAseq analysis of the hypothalamic tissue from Prader-Willi syndrome patients, which are known to be hyperphagic and morbidly obese, revealed the enrichment of microglial genes, suggesting a role of microglia in this disease [49].

## 3. Hippocampus

The hippocampus is a part of the limbic system involved in the consolidation of information from short- to long-term memory, as well as spatial memory. The subfield nomenclature varies among authors; however, the most widely accepted separation is the dentate gyrus (DG) and the so-called hippocampus proper, or cornu ammonis (CA), differentiated into three fields (CA1-3). The DG is suggested to be involved in the formation of memories, while the CA forms a neural circuit, transferring input and output signals between the cortex and DG, as well as between its three subregions. 

The effect of obesity on microglial neuroinflammation in the hippocampus has been studied broadly. Obesity is shown to be highly associated with changes in hippocampal function that could ultimately result in cognitive decline and dementia [50]. Both with short- and long-term DIO, microglial cells in the whole hippocampus showed an increased gene expression of the microglial activation markers Iba-1, Cd11b and CD45+, as well as MHC II [34,51,52,53]. Moreover, this effect seems to be age-dependent. Aging and obesity seem to exert a synergistic effect on microglial activation in the hippocampus, as indicated by the exacerbated microglial activation (CD68+) and enrichment of microglial proinflammatory genes, compared to aged or DIO groups alone [54,55]. Impairment in antioxidant production and reactive oxygen species (ROS) detoxification in obese animals exacerbated the neuroinflammation in the hippocampus, as evaluated by increased microglial activation [56]. These data point to a possible mechanism for the progression of cognitive impairment in obesity, as aging is associated with increased sensitivity to oxidative stress, which is known to affect cognition.

Different factors can influence microglial activation in obesity by exerting protective or deleterious effects on neuroinflammation. Exposing obese animals to alcohol further increased the hippocampal microglial activation, together with a decrease in the microglial cell number, as well as a loss of the morphological integrity [57]. Ultimately, such changes will result in a decreased inflammatory capacity.

Hippocampal microglial activation was also increased in db/db mice (genetically obese and leptin receptor-deficient), as demonstrated by the increased Iba-1+ cell numbers, as well as higher MHC II immunoreactivity and Il1b cytokine production, when compared to control animals [58]. This trend could be reversed fully by treadmill training, suggesting that exercise attenuates microglial cell activation in a leptin receptor-independent manner. Interferon regulatory factor 2 binding protein 2 (irf2bp2) is a key regulator of microglial polarization, thus regulating the innate immune response. A study showed that HFD-induced microglial activation (as seen in increased Iba1 expression) was further enhanced in Irf2bp2-deficient mice [34]. 

However, others failed to reproduce this hippocampal microglial activation by an obesogenic diet, reporting no differences in the microglial process length, Iba-1+ immunoreactivity and/or cell number [41,59,60]. One group even observed a decrease in the Iba-1+ area fraction in hippocampi from rats fed a hypercaloric diet, compared to control animals [61]. They reported no signs of gliosis in obese animals and suggested that obesity-associated cognitive decline is not solely driven by an increased fat consumption per se but is a result of a complex interplay of dietary composition and caloric intake, as well as other factors like age, physical activity and stress. 

### 3.1. CA1-CA3

Most of the observations mentioned above for obesity-induced microglial activation in the whole hippocampus also hold for the CA1 region. A robust increase in microglial activation (as seen in the morphological changes of a decreased process number, increased cell body area and increased phagocytic capacity) was found in the CA1 of obese animals, but no changes were observed in the microglial cell number [62,63,64,65,66]. Loss of the functional Toll-like-receptor 4 (TLR4, receptor involved in the lipopolysaccharide response) abolished CA1 microglial activation in obese animals and restored their morphology to a resting state [62]. Moreover, hypercaloric microglial activation in the CA1 was attenuated by long-term pre-, pro- and symbiotic treatments, suggesting a beneficial effect of pre-, pro- and symbiotics on microglial immunometabolism in obesity [63]. However, the exact mechanism behind this effect is yet unknown. One study reported that a pioglitazone (PPARg agonist) treatment reduced HFD-induced microglial activation in the CA1 and restored microglial cells to the control conditions [64]. Some effects seem to be more specific. A cafeteria diet increased Iba-1+ cell density in the CA1 region, which could be abolished by a reversal to the control diet [20]. However, an evaluation of the effect of the cafeteria diet on the microglial function in the whole hippocampal tissue, as well as the DG microglial cell density, showed no differences, suggesting a CA1 site-specific effect [20,67]. 

Both young and aged rats on an obesogenic diet presented with morphological changes in CA3 indicative of microglial activation. However, aged obese animals exhibited a significantly higher percentage of activated microglia, compared to young obese animals [68], suggesting that aging enhances the inflammatory response in a hypercaloric environment. 

### 3.2. DG

Obesity caused robust DG microglial activation with increased microglial cell numbers in the hilar region when compared to controls [59,65,69]. This microglial activation persisted in the molecular and granule dentate cell layers when animals were chronically exposed to an obesogenic diet. However, a reversal back to the control diet resulted in a full recovery of the microglial control state, indicating that the obesity-induced activation of microglia in the DG is reversible [70]. Another effective way of reducing microglial activation in the DG is exercise, as shown by the reduced Iba-1+ overexpression in obese animals subjected to treadmill running [71]. Remarkably, blueberry consumption has been shown to be a potent protector against microglial activation in the DG, as it significantly decreased the Iba-1+ immunoreactivity in obese animals [72]. 

### 3.3. Hippocampus and Alzheimer’s Disease in Obesity

Aging is associated with hippocampal impairment, which leads to short-term memory loss often also observed in neurodegenerative conditions like Alzheimer’s disease (AD). Rising evidence suggests a link between obesity (and its underlining comorbidities) and AD [73]. Microglial cells have been suggested to play a key role in the progression of AD, as amyloid beta can activate microglial cells and, thus, lead to chronic neuroinflammation with detrimental effects on the neuronal survival [74]. Obesity seems to further promote this amyloid beta-induced microglial activation, which would ultimately worsen AD outcomes [75,76,77]. Interestingly, a study in an AD model showed that in young 3xTg-AD female mice, microglial activation was increased following the HFD intake; however, no differences between the control and HFD-fed animals were observed in early and late middle-aged females [78]. Similar observations were made in male 3xTg-AD mice chronically fed HFD, where the microglial activation was increased in young, but was not changed in aged, obese animals [75]. These data suggest that, in combination with a hypercaloric challenge, the AD pathology progression is accelerated. However, others failed to reproduce these results and reported no effect of the hypercaloric diet on microglial activation in mice with cerebral amyloid angiopathy (CAA) [79]. The observed effect in CAA mice was limited to the CA1 region of the hippocampus, which could be a possible explanation for the lack of effect of obesity in AD reported earlier in this article on the whole hippocampal tissue. It is worth mentioning that, although many animal models have been developed for the study of AD, currently, none fully represents the pathology of AD [80]. 

## 4. Cerebral Cortex

The cerebral cortex is the outer layer of the brain and is known to be the largest site of neural integration in the CNS. The cortex is important for a well-functioning memory, language, perception and consciousness. Many aspects of its functioning have been studied in relation to obesity, aiming to decipher the mechanism of obesity-associated cognitive decline [50]. It has been proposed that microglial activation contributes to this cognitive impairment [50]. Only a few studies have reported microglial activation in the cortex of animals fed an obesogenic diet, which could be reversed by exercise (treadmill running) or treatment with an acetylcholine esterase inhibitor [81,82]. However, most studies failed to report the cortical activation of microglia in obesity [22,53,60,67,83,84,85,86], suggesting that an obesogenic diet does not have major effects on the microglial innate immunity in the cortex. This observation was also confirmed in human cortical tissue, with no differences observed between the obese (BMI > 30) and control (BMI < 25) subjects [24]. It seems, however, that in other pathological conditions, obesity does exacerbate the microglial activation in the cortex. An obesogenic environment resulted in an increased microglial activation during TBI compared to a TBI alone [45]. Moreover, chronic exposure to an obesogenic diet resulted in a higher microglial activation following infarction, induced by a 30-minute middle cerebral artery occlusion (MCAo) [87]. This microglial activation is likely due to ischemic damage; however, it is worsened in obesity, suggesting a priming effect of chronic hypercaloric intake. Interestingly, a study in AD mice (APP/E4) found sex-dependent differences in cortical microglial cells during obesity. In females, a decrease in Iba-1+ microglial cell coverage was found around plaques in animals fed HFD, mainly due to a decreased number of processes in contact with the plaque [88]. Since these differences were observed only in females, this suggests a sex-dependent effect. 

## 5. Striatum

The striatum is a nucleus in the basal ganglia primarily involved in the motor and reward systems. In obese humans, the striatum has also been implicated in food addiction [89]. However, striatal inflammation and microglial activation in obesity are poorly understood. A few studies of HFD-fed rodents showed no effects on the microglial activation status in the striatum, as no differences in the microglial cell number or morphological changes were observed [81,90]. Moreover, contrary to the observations in the cortex, obesity did not exacerbate the microglial activation following MCAo in the striatum [87]. Age seems to have an effect on microglial activation in obesity in the striatum, as increased microglial activation and the phagocytic capacity marker CD68+ were observed in aged obese mice [91]. The chronic feeding of mice with a cafeteria diet—the closest current equivalent to a human high intake of processed, calorie-dense food—was shown to result in microglial activation both in the nucleus accumbens (NAc) core and shell, located in the striatum [92]. This response was only observed following a chronic cafeteria diet intake, but intermittent exposure had no effect on the microglial activation. The NAc plays a key role in the motivation and reward system and has been linked to the addiction-like mechanisms taking place in obesity [93]. Therefore, studies focused more specifically on NAc alone, compared to the overall striatal neuroinflammation, might give a better indication of the microglial status in obesity. 

## 6. Other Brain Regions

Some studies have also reported the obesity-induced microglial status in other brain regions with no direct involvement in food intake, like the amygdala, cerebellum and corpus callosum. Chronic HFD feeding resulted in increased Iba-1+ immunoreactivity in the amygdala [59]. However, a short-term diet failed to reproduce these results [52]. In the cerebellum, no clear links were observed. An evaluation of the microglial morphology showed no differences between the control and obesogenic diet conditions, which was supported by an evaluation of the Cd11b gene expression [22,84]. Chronic obesity seems to increase the microglial activity in the corpus callosum of HFD-fed male mice, while no differences were observed in females [45]. This observation further supports the idea that microglial cells have sex-specific differences in the predisposition towards obesity. 

## 7. Effect of Overnutrition on Microglial Function in the pre- and Neonatal Stages of Life

Maternal obesity has been shown to be associated with adverse health outcomes in the offspring, some of which are an increased risk for the development of obesity, type 2 diabetes and cognitive dysfunction [94,95]. Moreover, neonatal overfeeding is associated with the development of obesity and insulin resistance [96]. Reducing the litter size in neonate rodents to three or four pups, compared to control size litters (10–12 pups), is a well-established technique to study neonatal overnutrition [97,98]. Both of these paradigms indicate that early overnutrition can have long-term effects on the energy homeostasis and significantly increase the susceptibility to obesity. 

In whole hypothalamic tissue, the protein levels of the Iba1 microglial marker showed no effect of the sex or overnutrition status at postnatal day (PND) 10 or 50. However at PND150, in both sexes, the overall hypothalamic Iba1 levels were decreased in response to neonatal overnutrition [99]. In another study, no differences in the Iba-1+ cell number and density were observed in males and females in any hypothalamic region at PND7 [100]. 

Neonatal overnutrition has mostly been studied in male animals. Neonatal overnutrition has been shown to lead to increased microglial activation (MHC II+) in adulthood in the VMH, optic chiasm and ME of rats on a standard chow diet [101]. A similar trend was observed in the cerebellum; however, it did not reach significance. The Iba-1+ microglial cell numbers were also increased in the PVN of neonatal-overfed male animals at PND14 and in adulthood. In females, neonatal overfeeding seemed to increase the microglial number only in the PVN in adulthood [100,102], although this effect was less pronounced compared to males. No differences were observed in adulthood and at PND14 in the Iba-1+ cell number or density in the ARC between control and neonatal overfed rats [99,100,103]. Interestingly, a short hypercaloric challenge (three-day HFD) increased the microglial cell number and density in the PVN of control litter rats but reduced the microglial number in overfed litter rats [102]. Thaler et al. showed that, in adult animals, the early inflammatory response to a hypercaloric challenge is an adaptive response [10]. This suggests that the lack of microglial activation response in neonatal overfed animals is due to a chronically altered microglial phenotype and impaired inflammatory reaction. 

A study on the effect of chronic maternal hypercaloric exposure in nonhuman primates showed an increased Iba-1+ microglial cell density and area in the ARC at gestational day 130 (early third trimester), suggesting that a hypercaloric environment has an effect on the microglial function not only at the neonatal stage and later but, also, during prenatal development [104]. 

Neonatal overnutrition was also shown to have effects beyond the hypothalamus in memory-associated brain regions like the hippocampal CA1, CA3 and DG. The maternal intake of a high-saturated-fat diet (SFD) or high-trans fat diet (TFD) was shown to have a potent effect on microglial activation in the offspring at PND1, as evidenced by the increased Cd11b mRNA expression in the hippocampus, both in males and females [105]. A detailed study of the Iba1 protein levels in the different hippocampal regions (CA1, CA3 and DG) of the offspring of SFD- and TFD-fed dams in the adult stage showed a significant increase in the SFD groups, compared to the TFD and control groups. Moreover, following an immune challenge with lipopolysaccharide (LPS), both the HFD and TFD groups showed increased Iba1 levels in all three regions, when compared to the controls [105]. Neonatal overfeeding also exerted an acute effect by increasing the microglial number and density in CA1 at PND14 when compared to the control and PND7 [106]. In the CA3 region, there was a trend for neonatal overnutrition suppressing the microglial numbers, while the density increased at PND14. In the DG, the microglial numbers and density increased with age in neonatally overfed animals in the subgranular/granular zone, increasing significantly from PND7 to PND14. There was a trend for an increased density in the hilar region of the DG [106]. No long-term effects of neonatal overnutrition were found in the microglial number or density in the CA1 and CA3 [106]. However, neonatal overfeeding increased the susceptibility to an immune challenge in these regions, with an increased microglial cell number in both regions following an intraperitoneal LPS injection compared to the controls [107]. In yet another study, the microglial cell number was increased in the hilar and subgranular/granular regions of the DG in neonatal overfed animals compared to the control litter. Moreover, neonatal overfeeding increased the microglial density in all the DG regions (hilar, subgranular/granular and molecular) when compared to control-fed rats under basal and LPS conditions [107]. 

In the cortex, neonatal overfeeding has been shown to acutely increase microglial activation, following hypoxic-ischemic (HI) brain injury, compared to lean HI animals at PND7 [108]. The hyperactivation of c-Jun amino-terminal kinase (JNK) activity has been shown to play a role in ischemic brain injury and obesity [109], and 76% of activated microglial cells expressed p-JNK against only 5% in the resting microglia [108]. The inhibition of JNK caused a significant reduction of the HI-induced microglial activation in neonatal overfed pups, with no effect in the lean HI group, suggesting a possible mechanism involved in microglial activation in neonatal overnutrition [108]. This effect for cortical microglial activation was also observed at PND1 following a maternal hypercaloric intake [105]. In another study, there was a significant increase in the microglial cell number at PND14 following a maternal hypercaloric intake, which was further exacerbated by a HI injury [110]. 

Prenatal air pollution has been shown to prime microglial cells long-term, resulting in an exacerbated inflammatory response following hypercaloric exposure in adulthood. Following a hypercaloric diet, adult males that received prenatal diesel exposure (DE) had an increased microglial CD11b mRNA expression in the hippocampus when compared to prenatal filtered air (FA) mice [111]. Interestingly, no differences were observed in the Cd11b mRNA expression in the hippocampal tissue from females [111]. An evaluation of the regional differences showed a greater Iba-1+ density in the DE hypercaloric group compared to the FA hypercaloric group and the DE control group in the CA1 and DG, while there was an overall greater Iba-1+ density in the hypercaloric group, regardless of the prenatal treatment [112]. A similar trend for increased microglial density was observed in the hypothalamus and amygdala from males in the hypercaloric DE group compared to all the other groups. In females, an increased expression of Iba-1+ was observed only in the hypercaloric DE group in the hypothalamus [112]. No effect from the diet or prenatal air treatment was observed in the CA3 region, either in males or females [112].

Taken together, these data suggest that a hypercaloric environment in the early developmental stages of life has a long-term deleterious effect on the microglial function. Additional factors like prenatal air exposure could further exacerbate this microglial activation. Moreover, pre- and neonatal overnutrition show sex- and age-dependent effects on gliosis. Figure 1 (located at the end of the results section) illustrates the main findings in microglial brain heterogeneity in obesity.

## 8. Conclusions

Microglial cells play a key role in obesity-associated neuroinflammation. The activation status of the microglia has been found to differ during obesity, depending on the brain region. The most pronounced changes are found in the hypothalamus, as the core brain region for the control of food intake and energy expenditure. However, a hypercaloric environment has been shown to result in microgliosis in other brain regions as well. Hippocampal HFD-induced microglial activation could play a key role in obesity-associated cognitive decline. Overnutrition has an effect on the microglial activation not only in the adult stage of life but, also, during the pre- and neonatal periods, which may result in a long-term priming of the microglial cells. Additionally, microglial cells show sex- and age-dependent differences in their activation status and response to an obesogenic environment. Although most studies focused on male rodents, several studies suggest that females have a protective mechanism against obesity-induced microglial activation. Further studies should be aimed at understanding the mechanisms behind this protective function in females. Moreover, future research should aim to address the microglial heterogeneity to give a better understanding of the disease progression in the different brain regions, as well as the effects of age and sex. 

## Figures and Tables

**Figure 1 ijms-22-03141-f001:**
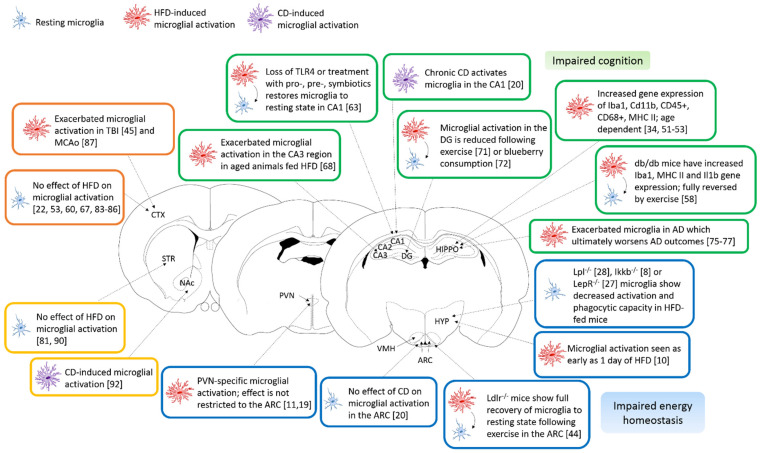
Microglial brain region heterogeneity in obesity. Schematic representation of the main findings of this review using mouse coronal brain slices to illustrate the regions of interest. Resting microglial cells are represented in blue, HFD-activated microglia in red and CD-activated microglia are in purple. The main findings for the hypothalamus are presented in the blue boxes, for the hippocampus, in green boxes, for the cortex, in orange boxes and, for the striatum, in the yellow boxes. Abbreviations: AD—Alzheimer’s Disease, ARC—Arcuate nucleus, CA1-3—Cornu Ammonis, CD—Cafeteria diet, CTX—Cerebral cortex, DG—Dentate gyrus, HFD—high-fat diet, HIPPO—hippocampus, HYP—hypothalamus, MCAo—Middle cerebral artery occlusion, NAc—Nucleus accumbens, PVN—Paraventricular nucleus, TBI—Traumatic brain injury and VMH—Ventromedial hypothalamic nucleus.

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
