# Peer review of "Mapping of Microglial Brain Region, Sex and Age Heterogeneity in Obesity"

_ijms, 2021, doi:10.3390/ijms22063141_

Round 1

Reviewer 1 Report

The current review deal with the impact of microglial innate immunity in obesity, more interesting, authors analyze the difference of this effect among different regions, genders, and ages. The review has a clear structure, separated according to the different brain regions evaluated, and summarizes the most important phenotypic differences of microglial innate immunity in obesity obtained from experimental research and elucidates it in a nice figure.  It is interesting for a board readership, especially for scientists dealing with metabolic syndrome and neuropathy.  I have only one suggestion: The authors divide the review by different brain regions, which are affected. I recommend adding a short table or conclusion at the end of each paragraph where the main literature is listed.

Author Response

We would like to thank the reviewer for taking the time to read and critically evaluate our work.  We are grateful for his/her kind words and positive response to our work. We have taken into consideration the suggestion of the reviewer and propose to include the major findings in this review article in the summary figure in the form of citations to improve the readability and traceability for the readers. We have replaced the figure with the adapted version with citations for each section (shown on the figure in brackets, e.g. [1], etc.) at line 503.

Reviewer 2 Report

The authors develop a review focusing on obesity related to changes in microglia. The subject is very interesting and relevant in science. To my understanding, the manuscript is well-developed, only being able to change the structure of the topics.

- CA1 and 3 could be one topic

- Since there are several diseases related to obesity, why only an Alzheimer's topic? This topic should be in the Hippocampus topic.

Author Response

We would like to thank the reviewer for his/her kind words and positive response to our work. We have taken into consideration the suggestions of the reviewer and have merged the CA1 and CA3 categories in one topic (line 297). We have removed the topic for Alzheimer’s disease and have left it as a separate paragraph (due to the substantial research performed on this topic) at the end of the Hippocampus section.

Reviewer 3 Report

This is a helpful review and adds needed clarification to the developing field of obesity mechanisms. There are several language corrections needed. For example , line 161 has a comma splice and elsewhere in the text. Lines 214 and 219 outcome not outcomes. Define other regions in text for better understanding ( CA1, CA3, DG). What are MCAo (line 392), and LPSip (line 472). Line 422 is awkward. Reference 88 is incomplete. I recommend a careful reading of the entire text to look for other areas that might need corrections. 

Author Response

We would like to thank the reviewer for his/her nice words and for pointing out the language corrections necessary to improve the review article. We have introduced a comma in line 161. We have corrected the word “outcome” in lines 214 and 219. The hippocampal brain regions have been defined in the text, starting at line 254. We have defined the abbreviations “MCAo” (line 371) and “LPS” (line 457) earlier in the text. We have replaced the abbreviation “i.p.” (previously line 472) with “intraperitoneal injection” (currently line 467). We have improved line 422 (currently line 418) and corrected reference 88 (line 741). We have read the entire work and introduced some final minor changes – we have introduced a comma (lines 38, 79, 108, 112, 157, 175, 301, 346), a hyphen (lines 97, 199, 209) and have made minor textual corrections to improve readability (lines 210, 211, 302, 505-506, 516).